# Knee joint loadings are related to tibial torsional alignments in people with radiographic medial knee osteoarthritis

**Chen Huang** [1], **Ping Keung Chan**[2], **Kwong Yuen Chiu**[2], **Chun Hoi Yan**[2], **Desmond Shun Shing Yeung**[3], **Christopher Wai Keung Lai** [4], **Siu Ngor Fu**[1]*

**1** Department of Rehabilitation Sciences, The Hong Kong Polytechnic University, Hong Kong, China, **2** Department of Orthopaedics and Traumatology, Queen Mary Hospital, The University of Hong Kong, Hong Kong, China, **3** Physiotherapy Department, MacLehose Medical Rehabilitation Centre, Hong Kong, China, **4** Health and Social Sciences Cluster, Singapore Institute of Technology, Singapore, Singapore

* amy.fu@polyu.edu.hk

## Abstract

Torsional malalignment was detected in subjects with medial knee osteoarthritis (KOA) but few studies have reported the effect of torsional deformity on knee joint loads during walking. Therefore, this study examined the relationships between lower limb torsional alignments and knee joint loads during gait in people with symptomatic medial KOA using cross-sectional study design. Lower limb alignments including tibial torsion, tibiofemoral rotation and varus/valgus alignments in standing were measured by EOS low-dose bi-planar x-ray system in 47 subjects with mild or moderate KOA. The external knee adduction moment (KAM), flexion moment (KFM) and the KAM index which was defined as (KAM/ (KAM+KFM)*100) during walking were analyzed using a motion analysis system so as to estimate the knee loads. Results revealed externaltibial torsion was positively associated with KAM in subjects with moderate KOA (r = 0.59, p = 0.02) but not in subjects with mild KOA. On the contrary, significant association was found between knee varus/valgus alignment and KAM in the mild KOA group (r = 0.58, p<0.001) and a sign of association in the moderate KOA group (r = 0.47, p = 0.08). We concluded tibial torsion and knee varus/valgus mal-alignments would be associated with joint loading in subjects with moderate medial KOA during walking. Radiographic severity might need to be considered when using gait modification as a rehabilitation strategy for this condition.

## Introduction

Knee osteoarthritis (KOA) is a common degenerative joint disorder in seniors. Among all KOA categories, medial compartment KOA has accounted for 67.8% in the eastern population [1] and 85.4% in the western population [2]. The more vulnerability of developing KOA in the medial compartment was linked to the higher joint load on the medial than the lateral part of knee [3]. Joint malalignment, besides being a risk factor associated with knee joint load

**Data Availability Statement:** All relevant data are within the manuscript and its Supporting Information files.

**Funding:** The authors received no specific funding for this work.

**Competing interests:** The authors have declared that no competing interests exist.

distribution during weight-bearing activities [4], was also regarded as a predictor to articular cartilage loss [5] and KOA progression [6].

Tibial torsion, defined as the rotational alignment between tibial plateau and malleoli, was altered in people with KOA [7], such that they had less external tibial torsion than people without KOA [7–9]. Besides external tibial torsion being smaller, these subjects also would have an increased varus malalignment as the disease progressed in the medial knee compartment [8]. A cadaveric study has reported both excessive internal tibial torsion of over 20° and more external tibial torsion would increase the medial compartment knee joint contact pressure [10]. These findings suggested either a reduced or increased external tibial torsion would induce a change in joint contact pressure. In subjects with end-stage medial KOA and internal tibial torsion malalignment, their second peak of KAM during walking was higher than those without any torsional malalignment and the normal controls [11]. These suggested the joint contact force and mechanical load distribution of the knee might be related to tibial torsion. If this relationship did exist, then knowing how these two factors behave at the early stance phase of gait would be important because the peak KAM during stance phase of walking was a strong predictor for the long-term medial-to-lateral cartilage erosion [12]. The dynamic internal tibiofemoral rotation during the early stance phase was reduced in people with medial compartment KOA [13, 14], and the reduction became more evident when the condition had progressed [15]. It is therefore necessary to measure this rotation and its effect on joint loads during walking.

Recently, low dose bi-planar x-ray scanner has become available to measure the torsional alignments, by which scanning was conducted in a weight-bearing condition [16, 17]. This study aimed to use this new technology for capturing the joint alignment in a weight bearing condition, so as to investigate the association between lower limb alignments and knee joint loads in people with medial compartment KOA. We hypothesized that external tibial torsion and tibiofemoral rotation were associated with the joint loads during walking. More specifically, reduced external tibial torsion and internal tibiofemoral rotation would be associated with greater external knee moments in the stance phase. By exploring such associations, we could then develop a better understanding on how torsional alignment would affect joint loading in subjects with KOA.

## Methods

### Study design

This is a cross-sectional observational study.

### Subjects

Subjects with KOA attending the orthopaedic clinic of a local hospital were invited to join the study if they met the following criteria: 1) aged between 50 and 80 years, 2) had radiographic evidence of KOA in the medial compartment of tibiofemoral joint with Kellgren-Lawrence (KL) grading of less than 4, 3) had a minimum pain score of 2 on an 11-point visual analog scale (VAS) in the past month while walking [18]. Subjects would be excluded if they had any of the followings: 1) more osteophytes in the lateral than the medial knee compartment, 2) intra-articular injection within the past 6 months, 3) rheumatoid arthritis, 4) history of surgery to either knee joint, 5) other muscular, joint or neurological conditions influencing lower limb function, 6) low back, hip, ankle or foot pain of more than 3 on VAS, 7) unable to walk without assistance and, 8) body mass index (BMI) >36 kg/m$^2$.

**Table 1. Demographic information of subjects.**

|  | All (n = 47) | Mild (n = 30) | Moderate (n = 17) |
|---|---|---|---|
| Age (year) | 62.1 ± 6.0 | 60.7 ± 5.8 | 64.4 ±5.8 * |
| Gender (Female/male) | 37/10 | 27/3 | 10/7 * |
| Height (m) | 1.6 ± 8.6 | 1.6 ±6.6 | 1.6± 11.6 |
| Weight (kg) | 65.4 ± 11.5 | 62.8 ± 9.5 | 70.0 ± 13.5 |
| BMI (kg/m$^2$) | 26.4 ± 3.6 | 25.4 ± 3.4 | 27.8 ± 3.4 * |
| Pain (VAS) | 5.12 ± 1.8 | 4.8 ±1.7 | 5.8 ± 1.7 |
| Walking speed (m/s) | 1.0 ± 0.2 | 1.1 ±0.2 | 0.9± 0.2* |

* Difference between mild and moderate group p<0.05.

Diagnosis and grading of the OA severity by KL scaling system were determined by experienced orthopedic surgeons. The severity of KOA was categorized as mild (KL grade 1 and 2) and moderate (KL grade 3) [19] and these categories would be included in the analysis.

Forty-seven subjects were recruited. Demographic information of the subjects was shown in Table 1. In general, subjects aged between 50 and 77 years old and 79% of them were female. The severity of KOA was classified as mild in 30 subjects and moderate in 17. The sample size was reached according to the availability of eligible patient in the study period.

This study followed the principle of the Declaration of Helsinki and was approved by the Human Subjects Ethics Sub-committee, Department of Rehabilitation Sciences, The Hong Kong Polytechnic University (Ref. HSEARS20170406003). All subjects gave their written informed consent before being tested.

## Outcome measurement

**Joint alignment.** Joint alignment during stance was examined with a low-dose bi-planar X-ray imaging system (EOS imaging, Paris, France) [20] by an experienced radiographer. Subjects were asked to stand in the center of the testing gantry with their right leg shifted forward for around 4 cm to ensure clear bony structure recognition. A three-dimensional reconstruction of the lower limbs was performed with sterEOS (Version 1.6, EOS imaging, Paris, France). Anatomical reference points were identified on both the sagittal and coronal planes including femoral head, neck, greater and lesser trochanters, intercondylar notch, lateral and medial femoral condyles, tibial spine, lateral and medial tibial plateau, distal tibial articular surface and medial malleolus. Bony contours were adjusted according to the EOS guidelines by the assessor.

Knee alignment angles were computed by the sterEOS software. Tibial torsion angle was measured by drawing from the posterior tibial plateau tangent axis to the bi-malleolar axis (Fig 1A), whereas tibiofemoral rotation angle was measured from the posterior femoral bi-condylar axis to the posterior tibial plateau tangent axis (Fig 1B). A clockwise value was defined as external and a counter clockwise value was internal rotation for the right leg while the reverse applied for the left leg. Knee varus/valgus was measured as the angle between the longitudinal axes of femur and tibia. Reliability test was conducted in 6 subjects and excellent inter-day reliability was obtained for knee varus angle (ICC = 0.99, p <0.001), tibial torsion (ICC = 0.93, p<0.001) and tibiofemoral rotation (ICC = 0.95, p<0.001).

**Joint loads during gait.** Knee loads were estimated using a motion analysis system that comprised eight cameras (MX T40, Vicon, Oxford, UK) and two floor-mounted force plates (Kistler Group, Winterthur, Switzerland). The sampling rates were 1000Hz for kinematic data and 100Hz for kinetic data. The two data sets were recorded and synchronized. Reflective skin

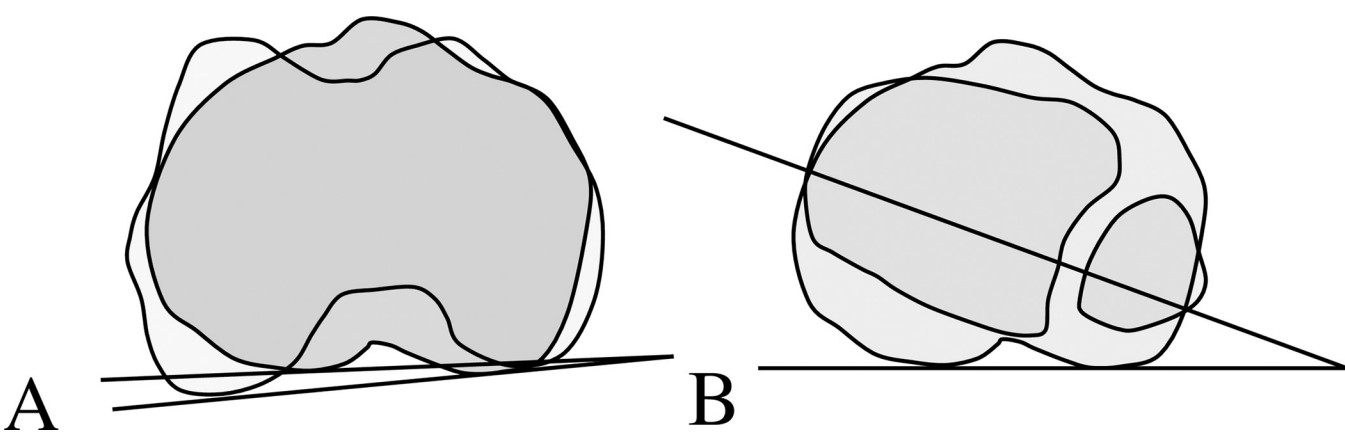

**Fig 1.** Diagram of torsional alignments measurement (A: tibiofemoral rotation, B: tibial torsion).

markers were applied according to the guidelines of standard Lower limb Plug-In-Gait marker set [21]. Subjects were instructed to walk unshod and without any assistive device on an 8-meter footpath in self-selected speed. A data set was collected during static standing for lower limb model building. Practice trials were given to the subjects for them to acquaint with the testing procedures. Data collection would start after the practice trials and a minimum of five successful walking trials in each leg with clean foot strike from heel-strike to toe-off on the force plates would be recorded [22].

External knee joint moments were calculated with Vicon Nexus software (Version 2.5, Oxford, UK) using Lower limb Plug-In-Gait model. Gait events of heel strike and toe-off were identified when the magnitude of the force plate was above and below 10N, respectively. Peak KAM and KFM in this study were respectively defined as the maximum values of knee moment in the frontal and sagittal planes during the initial 50% of stance phase. They were normalized to body weight and reported in Nm/kg. A KAM index was calculated using the formula: (KAM/ KAM + KFM)*100, which represented the percentage of load sharing between the frontal and sagittal planes. All kinetic data were estimated by the average of five successful trials.

## Statistical analysis

Statistical analysis was conducted using SPSS (Version 23.0, IBM Corp., New York, US). Distribution of each variable was assessed by Shapiro-Wilk test. Demographic group differences of radiographic severity were assessed by Welch's t test for unequal variances. Homogeneity of variance was tested by Levene's test.

Analysis of covariance was used to estimate group difference on joint alignment controlling for age and gender, and on knee joint loads controlling for gender, pain intensity and speed. Correlations between demographic variables, alignments and kinetic outcomes were tested by Spearman's rho test. Partial correlation coefficient tests were conducted between knee alignments and joint loads controlling for gender and pain for all subjects. Sub-group analyses according to radiographic severity were also conducted. This study contained no missing data.

## Results

### Correlation between torsional alignments with knee kinetics

Table 2 showed the relationships between knee alignments and joint loads during the stance phase of walking. Tibial torsion demonstrated an insignificant association with KAM when

**Table 2. Correlation between alignments and knee joint loads in medial knee OA.**

| | Tibiofemoral rotation | | Tibial torsion | | Knee varus angle | |
|---|---|---|---|---|---|---|
| | r | p value | r | p value | r | p value |
| All subjects | | | | | | |
| KAM | -0.11 | 0.45 | 0.06 | 0.69 | 0.58 | <0.001 |
| KFM | 0.07 | 0.66 | -0.02 | 0.89 | -0.07 | 0.65 |
| KAM index | -0.05 | 0.76 | 0.14 | 0.38 | 0.31 | 0.04 |
| Subjects with Mild knee OA | | | | | | |
| KAM | -0.03 | 0.88 | -0.19 | 0.34 | 0.68 | <0.001 |
| KFM | 0.04 | 0.84 | 0.09 | 0.63 | -0.03 | 0.88 |
| KAM index | -0.01 | 0.97 | -0.07 | 0.73 | 0.35 | 0.07 |
| Subjects with moderate knee OA | | | | | | |
| KAM | -0.02 | 0.94 | 0.59 | 0.02 | 0.47 | 0.08 |
| KFM | 0.04 | 0.89 | -0.39 | 0.15 | -0.05 | 0.86 |
| KAM index | 0.00 | 1.00 | 0.56 | 0.03 | 0.25 | 0.38 |

*Controlling for gender and pain intensity.

analysis was done by combining both the moderate and mild OA groups (r = 0.07, p = 0.67; Fig 2A). However, sub-group analysis revealed a significant positive association between tibial torsion and KAM in the moderate KOA group (r = 0.59, p = 0.02; Fig 2B) but not with the mild KOA group (r = -0.19, p = 0.34; Fig 2C).

The tibiofemoral rotation and KAM were not inter-related with either combined or sub-group analyses. Besides, there was no relationship between joint alignments and KFM with the combined or sub-group analyses.

Knee varus angle and KAM were significantly associated with one another when analyzed for all the subjects (r = 0.58, p<0.001; Fig 2A). With sub-group analysis however, the association was only significant in the mild KOA group (r = 0.68, p<0.001, Fig 2B) but not for the moderate KOA group (r = 0.47, p = 0.08, Fig 2C).

## Group differences on joint alignments and joint loading

There was no significant difference in tibial torsion (p = 0.58), tibiofemoral rotation (p = 0.15) or knee varus angle (p = 0.10) between the mild and moderate KOA groups but the moderate group had significantly higher KAM than the mild group (p = 0.02). There was no significant group difference in KFM (p = 0.55) or KAM index (p = 0.16). Details could be found in Table 3.

## Discussion

This study investigated the associations between knee torsional alignments and kinetic properties during early stance in people with medial compartment KOA. Although the finding revealed an association between tibial torsion and KAM in subjects with moderate KOA, the results did not support our hypothesis that tibiofemoral rotation would affect external knee moments.

We adopted the tibial torsion and tibiofemroal rotation as measures of torsional alignments and the values obtained in this study were similar to those reported in the literature that tibial torsion was between 25˚ and 27˚ [16, 23, 24] whereas tibiofemoral rotation was between 2° and 4˚ [23, 25] which were reportedly measured by computed tomography in people with medial KOA. The fact that we had used a relatively new approach of EOS and the

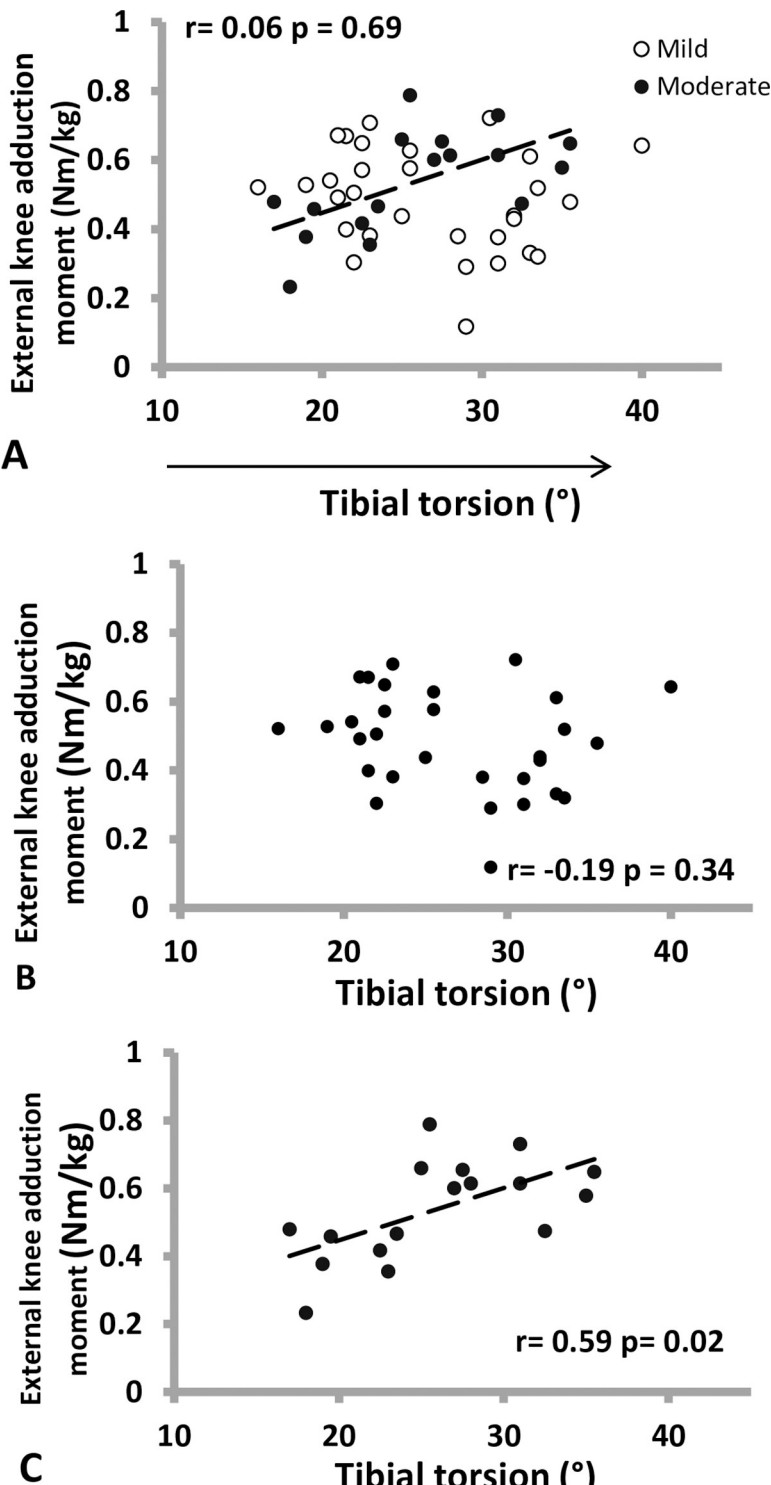

**Fig 2.** Scatter plots of external knee adduction moment and tibial torsion in different OA severity (A: all subjects, B: mild OA, C: moderate OA).

**Table 3. Alignment and knee joint loads in mild and moderate knee OA.**

|  | All | Mild | Moderate | P value |
|---|---|---|---|---|
| Alignment [a] |  |  |  |  |
| Tibial torsion (˚) | 26.5 ±5.8 | 26.8 ±5.8 | 25.9 ± 5.8 | 0.58 |
| Tibiofemoral rotation (˚) | 1.6 ± 8.1 | 4.2 ± 7.9 | -3.0 ± 6.2 | 0.15 |
| Knee varus angle (˚) | 6.7 ± 5.0 | 4.4 ± 3.1 | 10.7 ± 5.1 | 0.10 |
| Kinetics [b] |  |  |  |  |
| External knee adduction moment (Nm/kg) | 0.5 ± 0.2 | 0.5 ± 0.2 | 0.5 ± 0.2 | 0.02* |
| External knee flexion moment (Nm/kg) | 0.5 ± 0.3 | 0.5 ± 0.3 | 0.5 ± 0.2 | 0.55 |

a. Adjusting for gender, age.

b. Adjusting for gender, pain intensity and walking speed.

measurements were taken in a weight-bearing condition and generated similar values as those reports using computed tomography is suggestive that our method can be applied in future for weight-bearing activities to simulate more functional testing.

The main finding of this study was the association between tibial torsion and joint loading. More specifically, greater external tibial torsion was associated with larger KAM and KAMI in subjects with moderate KOA. This result indicated that subjects with more external tibial torsion were likely to have larger medial compartment joint load in subjects with moderate KOA. It was consistent with a previous cadaveric study by Kenawey et al. [10] which measured joint contact pressure directly. They found the joint contact pressure of knee medial compartment had a linear trend of increase within the range from 10˚ internal tibial torsion to 60˚ external tibial torsion [10]. In this study, the external tibial torsion ranged from 16˚ to 40˚ that fell into the range mentioned by Kenawey et al.; increase in external tibial torsion would associate with joint contact pressure of knee medial compartment within this range. Furthermore, increase in external tibial torsion was associated with higher KAM, and lower KAM was likely to be found in those with more internal tibial torsion. Although medial joint contact pressure began to increase after 10˚ of internal tibial torsion, the contact pressure at 20˚ of internal tibial torsion was still similar to that at neutral position [10]. This indicated a small internal tibial torsion of less than 20° might be biomechanically beneficial to people with moderate KOA. However, since none of our subjects had excessive internal tibial torsion, it cannot be ruled out that higher internal tibial torsion might increase the medial compartment load [26]. Also, tibial torsion was strongly associated with foot progression angle in the adult population [27]. Toe-out gait pattern was observed in subjects with greater external tibial torsion [27], and increased early stance KAM was observed in those walked with toe-out gait whereas the KAM would reduce in those with toe-in gait in subjects suffering from medial compartment KOA [28–30]. Therefore, these might explain the present findings that more external tibial torsion was associated with greater KAM. Further study is suggested to investigate the relationship between tibial torsion and foot progression angle.

Besides, the association between tibial torsion and KAM was only revealed in the moderate group but not in the mild KOA group. In subjects with moderate KOA, tibial torsion could have a stronger effect on the knee rotational compensatory mechanism [31, 32], which would help to shift the force transmission from the frontal to the sagittal plane that might reduce KAM.

There was no significant difference in tibial torsion between subjects with mild and moderate KOA. This was different to a previous report that people with advanced stage of KOA had more internal tibial torsion [8]. This difference might have a cultural origin as there was

habitual difference in the daily living between the subjects of our study and that of Yagi [8]. It echoed with the report of Tamari et al. in which larger internal tibial torsion in KOA was only found in Japanese rather than Caucasians [33]. The traditional Japanese sitting posture was kneeling on the floor with shank internally rotated which could drive the tibia to larger internal torsion as demonstrated in Yagi's study [8].

Originally, we thought tibiofemoral rotation would be related to knee joint load in people with KOA. However, our findings could not establish any relationship between tibiofemoral rotation and KAM or KFM. The dynamic internal tibiofemoral rotation toward the end of knee extension was described as the "screw-home mechanism", which has a functional importance to lock the knee and stabilize the joint [34]. This mechanism was altered in people with KOA [13] and even disappeared in those with advanced stage of KOA [35]. Absence of the "screw-home mechanism" could affect the joint stability thus increase the joint load. However, our results did not reveal tibiofemoral rotational alignment during static standing had any relationship with either KAM or KFM. It implied that the role of tibiofemoral rotation on joint loading under static standing and dynamic walking conditions was different, and this needs to be further explored in future studies.

It has been well reported KAM was directly related to varus deformity in people with KOA [36]. Biomechanically, larger varus deformity was related to longer level arm from the joint center on the frontal plane thus leading to greater KAM. We observed greater varus deformities in subjects with moderate KOA but were surprised to found that the association between knee varus angle and KAM had become weaker in this subject group, although the combined result of the two groups still revealed KAM to be strongly associated with knee varus angle. The reason for this finding remained unknown, and we postulated that it could be due to the influence of tibial torsion as discussed before, but the exact mechanism has yet to be explored. Notwithstanding that, correction of knee varus deformity was still regarded as the optimal management strategy to reduce KAM in people who had total knee replacement due to KOA [37].

There were a few limitations in this study. First, joint loading on the transverse plane was not reported. The reason we did not report the transverse joint loading was because of its minimal contribution at the stance phase [38] and its relatively low accuracy when captured using the motion analysis system [39]. Second, we did not include subjects with advanced KOA, who usually have severe joint malalignment. The excessive torsional malalignment would likely have a significant effect on knee biomechanics during gait. However, there is a technical difficulty to examine people with advanced KOA because of the massive osteophytes in their knee joints would obscure the bony contours in the bi-planar x-ray images rendering it difficult to identify the torsional angles.

In conclusion, this study found significant associations between external tibial torsion and early stance KAM in people with moderate medial compartment KOA. Tibial torsion angle should therefore be considered when designing biomechanical modification to decrease mechanical loadings of the knee in people with moderate KOA.

## Supporting information

**S1 Dataset.**
(XLSX)

## Acknowledgments

The authors thank Mrs. Harriet Ko for her assistance in data collection. We also want to thank Dr. Raymond Chung in giving statistical assistance and Prof. Gabriel Ng for advice in manuscript preparation.

## Author Contributions

**Conceptualization:** Chen Huang, Siu Ngor Fu.

**Data curation:** Desmond Shun Shing Yeung.

**Formal analysis:** Chen Huang, Siu Ngor Fu.

**Investigation:** Chen Huang, Ping Keung Chan, Chun Hoi Yan, Desmond Shun Shing Yeung, Christopher Wai Keung Lai.

**Methodology:** Christopher Wai Keung Lai, Siu Ngor Fu.

**Resources:** Ping Keung Chan, Kwong Yuen Chiu, Chun Hoi Yan, Desmond Shun Shing Yeung.

**Supervision:** Kwong Yuen Chiu.

**Writing – original draft:** Chen Huang.

**Writing – review & editing:** Chen Huang, Ping Keung Chan, Kwong Yuen Chiu, Chun Hoi Yan, Desmond Shun Shing Yeung, Christopher Wai Keung Lai, Siu Ngor Fu.

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
